# Efficient Communication Model for a Smart Parking System with Multiple Data Consumers

**T. Anusha *** and **M. Pushpalatha**

Department of Computing Technologies, SRM Institute of Science and Technology, Chennai 603203, India
* Correspondence: aa5293@srmist.edu.in

**Abstract:** A smart parking system (SPS) is an integral part of smart cities where Internet of Things (IoT) technology provides many innovative urban digital solutions. It offers hassle-free parking convenience to the city dwellers, metering facilities, and a revenue source for businesses, and it also protects the environment by cutting down drive-around emissions. The real-time availability information of parking slots and the duration of occupancy are valuable data utilized by multiple sectors such as parking management, charging electric vehicles (EV), car servicing, urban infrastructure planning, traffic regulation, etc. IPv6 wireless mesh networks are a good choice to implement a fail-safe, low-power and Internet protocol (IP)-based secure communication infrastructure for connecting heterogeneous IoT devices. In a smart parking lot, there could be a variety of local IoT devices that consume the occupancy data generated from the parking sensors. For instance, there could be a central parking management system, ticketing booths, display boards showing a count of free slots and color-coded lights indicating visual clues for vacancy. Apart from this, there are remote user applications that access occupancy data from browsers and mobile phones over the Internet. Both the types of data consumers need not collect their inputs from the cloud, as it is beneficial to offer local data within the network. Hence, an SPS with multiple data consumers needs an efficient communication model that provides reliable data transfers among producers and consumers while minimizing the overall energy consumption and data transit time. This paper explores different SPS communication models by varying the number of occupancy data collators, their positions, hybrid power cycles and data aggregation strategies. In addition, it proposes a concise data format for effective data dissemination. Based on the simulation studies, a multi-collator model along with a data superimposition technique is found to be the best for realizing an efficient smart parking system.

**Keywords:** smart parking system; data consumers; communication model; IPv6 Mesh; RPL; smart city; IoT

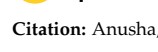



## 1. Introduction

Social, technological and economic factors contributed to the emergence of smart parking systems (SPS), and the recent advancements in electric and autonomous vehicles present a strong business case for intelligent parking services [1]. Currently, they are an urban requirement where users can search, navigate, reserve and pay for a free parking slot on a real-time basis. Countries across the world are turning to smart parking solutions for reducing traffic, minimize effort spent on parking, combat illegal parking, cutting down emissions as well as a business model to generate revenue. The global smart parking market is projected to grow at a compounded annual growth rate of 19.8% and is on its way to becoming a 16.3 billion dollar market in another six years [2]. As SPS matures, it fuels expansion of allied sectors such as sensor technology, Internet of Things (IoT) devices, communication access technologies, Machine to Machine (M2M) standards, smart city infrastructure projects and security solutions. As the roll out of 5G infrastructure facilitates real-time data availability with ultra low latency, IoT is excepted to realize its full potential [3].

All the literature on SPS concentrates on parking sensors that are data producers, access technologies and various software solutions. The data consumers, who access the generated parking data, are by default expected to be connected to cloud for their input. However, a smart city is analogous to the presence of heterogeneous IoT devices that consume data during M2M interactions [4]. Multiple data consumers are inevitable in an SPS, as various IoT devices are present in the parking lot. A workstation in the control room or a display device needs only local data. Whereas, an end user's mobile application may need more sophisticated data from a central cloud as it accesses the data over Internet. Hence, receiving data from the cloud may not be the best approach for on-site consumers because it takes up additional time in sending and receiving data through the Internet. A robust communication model is essential for establishing a quick, reliable communication between the data producers and consumers in a smart parking system.

Figure 1 shows the possible layout of a standalone parking lot equipped with different types of IoT devices.

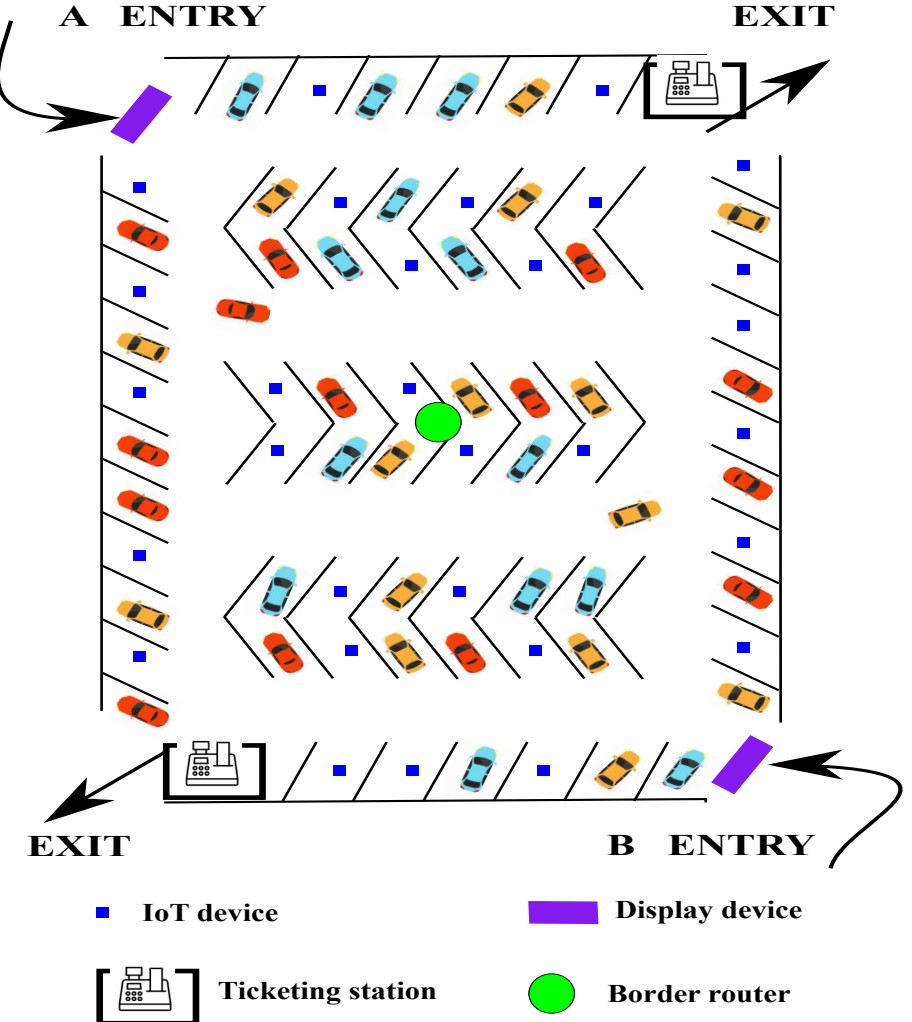

**Figure 1.** IoT infrastructure in a standalone parking lot.

IoT devices are installed at each parking space for gathering accurate data on availability, location and the duration of parking. These IoT devices are battery operated, simple to install and can last up to five or six years of operation without maintenance. The border router heads the mesh network and offers a global prefix for each device to equip them with a global IPv6 address. Parking availability data are locally consumed by the ticketing stations, which are present at the exit points and by the display screens, positioned at the entry points. As these devices depend only on the data collected from a standalone parking

lot, they are labeled as on-site data consumers. Consumers that require global collated data from multiple parking lots are off-site data consumers, who access the same over the Internet or cloud.

A survey by J. J. Barriga et al. found that an SPS is predominantly implemented using Zigbee networks in 60% of the studies followed by 25% with IEEE 802.15.4 [5]. However, data collection networks are not the best candidates for M2M communication between IoT devices. RPL [6] is the IPv6 routing protocol for low-power lossy networks, whose directed cyclic graph (DAG) formation is best suited for networks incorporating local data consumers. In RPL's storage mode, heads of subtrees store the routes to nodes that are underneath them and hence provide fail-safe communication paths between thr various nodes in the network. This paper evaluates different types of communication models for an SPS with multiple local data consumers, using the RPL routing protocol, in an IEEE 802.15.4 mesh network.

The next section briefly identifies various data consumers that are present in a smart parking system and categories them as either on-site or over-the-Internet type. Section 3 summarizes the related research works, and Section 4 elaborates on the different aspects of efficiency for a communication model. The further section evaluates the models, discusses their relevance and converges on an efficient multi-collator communication model.

## 2. Data Consumers in an SPS

A smart parking system is a complete digital platform that manages city-wide parking resources in real-time and provides multiple services to end users [7,8]. Starting from searching for a nearby available parking slot, booking a parking slot in advance, navigating to an available parking lot, charging electric vehicles at the booked slot, and predicting the availability for a specific time until payment for parking or charging is possible with a SPS. An smart parking lot utilizes various sensors for the accurate identification of empty slots, parking boundaries and the automated counting of the number of entries and exits. Apart from these sensors, it may have other IoT devices such as overhead LEDs as indicators for vacancy, LCD displays showing layout/availability statistics, buzzers or alarms for indicating wrong parking, automated gates that open after payment verification or license plate identification. These IoT devices need data from the sensors for their intended operations and are data consumers in a smart parking system.

A cloud-based SPS collates availability data from the sensors and sends them to the cloud for storage and processing. The application(s) on cloud servers provide relevant decisions and inputs to the data consumers. In such a system, it is required that the data consumers are connected directly to the cloud. This may not be an economical solution, as a direct Internet connection is required for all the consumers. For instance, an overhead LED light, showing the occupancy of a specific parking lot, just needs input from the respective parking sensor. Such a local scope does not need a cloud SPS. A mesh with any-to-any traffic support is most suitable. Multi-hop mesh networks are a cost-effective way of connecting heterogeneous IoT devices and providing Internet connectivity to all nodes in a network. There is no gateway device involved in an IPv6 mesh as there are no protocol translations, and all the communication is IP based. In addition, local data consumers can be given instructions within the network by the border router itself in a scaled-down centralized approach, reducing the round-trip time taken by the data. Data consumers can be classified as on-site or over-the-Internet, depending on the scope of the data consumed by them.

### 2.1. On-Site Data Consumers

On-site data consumers are IoT devices that work with the data generated from devices that are in its proximity. Figure 2a–d depict some of the IoT devices that are employed in individual smart parking lots. Smart LED bulbs are used to provide visual clues for drivers in a closed parking system that has less day light visibility. These smart LEDs can be connected to other IoT devices through WiFi or BlueTooth technology. Similarly,

smart alarms devices are also available and could be integrated to the central parking management system. The availability of systems on chips (SoC) supporting multiple radios in a single chip equips an IoT device to switch between different types of communication channels for device-to-device interaction. For instance, Qualcomm QCA4020 SoC provides intelligent multi-radio connectivity with WiFi, BlueTooth and 802.15.4 support [9].

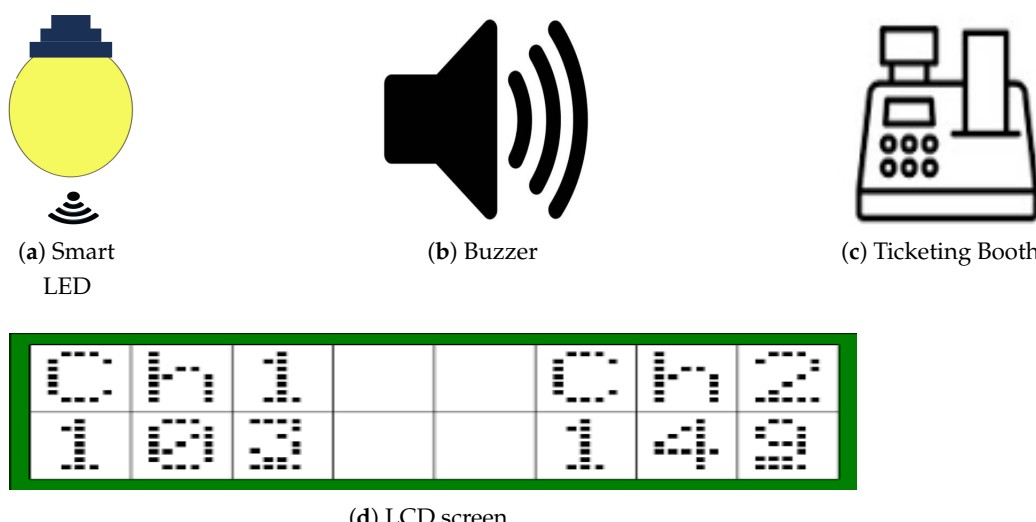

(**a**) Smart LED                                 (**b**) Buzzer                                 (**c**) Ticketing Booth

(**d**) LCD screen

**Figure 2.** Examples of on-site data consumers. (**a**) Smart bulb. (**b**) Buzzer for alarm. (**c**) Ticketing booth. (**d**) Display screen.

The ticketing booths could be a simple hand-held device with ticket or receipt-printing capability. They would need occupancy duration and timings for calculating relevant parking charges. They could also be a complex system, complete with an automated toll gate to allow passage for vehicles after verfication. Figure 3 shows a simple LCD screen display showing the aisle numbers and the respective numbers of lots that are vacant in them. Such a display screen, placed at the entrances of different levels, help users in a multi-level parking system. It could also be a complex system complete with a map in a very large parking lot.

| Parking Availability | |
|---|---|
| A1 | 0 0 1 |
| A2 | 0 2 5 |
| A3 | 1 1 7 |
| A4 | 0 5 4 |
| A5 | 0 0 7 |
| A6 | 2 5 5 |

**Figure 3.** A display screen placed in a smart parking lot, showing parking availability.

### 2.2. Over-the-Internet Data Consumers

Off-site data consumers are remote devices that access the parking lot occupancy data combined with other systems such as maps or payments. Figure 4a,b show mobile applications for booking parking lots or viewing parking lots available in a particular area. In contrast, Figure 5 presents a web page that provides a passive view of the availability information from the parking sensors. These are good examples for remote data consumers that need Internet access to receive data from an SPS. In an IPv6 mesh, the BR advertises a global prefix, and hence, the nodes become accessible over the Internet. The IoT devices are capable of hosting a web page, and hence, it can supply the occupancy data to a web browser upon an HTTP request. The example for such a web page is as shown in Figure 5, where a laptop is connected to the BR through a Serial Line Internet Protocol (SLIP). SLIP allows IP datagrams to be encapsulated and exchanged over serial ports. The web page can be accessed through the IPv6 address of the data collator.

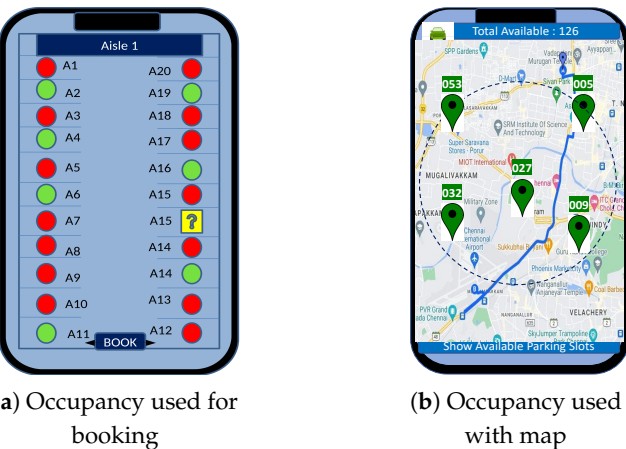

(**a**) Occupancy used for booking

(**b**) Occupancy used with map

**Figure 4.** Mobile application with multiple services. (**a**) Booking individual parking slot. (**b**) Searching for available parking slots in a map.

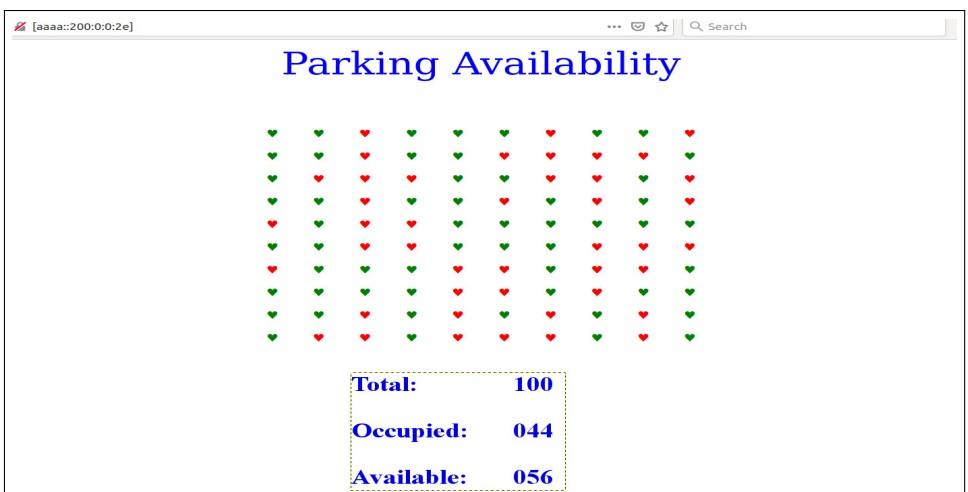

**Figure 5.** Over-the-Internet data consumer, a web browser showing parking occupancy.

Such direct BR to Internet connections work for a limited number of devices. However, if the number of simultaneous users increases, it is preferential to host the application in the cloud, as it provides elasticity in meeting user demand. In addition, web and mobile applications in the cloud have the privilege of collating data from multiple parking lots to show area-wide or city-wide car park availability. They can integrate the parking system data with other systems to offer seamless services.

## 3. Related Work in Literature

SPS systems are generally assisted or non-assisted where an assisted SPS allocates parking lots intelligently after considering various parameters such as the slot availability, user preference, closeness and traffic pattern of the route. This category strives to move forward toward autonomous vehicles and the navigation. A non-assisted SPS is a partially manual system where occupancy data are provided to end-users, and the actions are left to their decisions. Lin et al. classify SPS from another perspective based on the methods employed for information collection, the deployment technique of the system and their service dissemination model [8]. Paidi et al. show the gaps in deploying existing sensors and technologies for an open parking lot and the ways to design a robust multi-agent open parking lot [10]. Other works categorized parking systems based on the services offered by them such as parking reservation, guidance and crowdsourcing [11]. A survey by Fahim et al. identifies 12 different types of SPS systems depending on the technology used for sensing (Vision-based/GPS), communication (BlueTooth/WSN) or the learning models (ML/Fuzzy) employed [12]. In all these categories of SPS, a layered architecture is defined with the sensing layer at the bottom, the application layer at the top and a communication layer in between these two layers [13,14]. Al-Turjman et al. add one more layer for middleware to collate data from the sensors deployed at the parking lot [15].

The communication model of the available smart parking system perceives the sensing layer as a single unit where the sensors transmit the occupancy data to a central controller [16–18]. In rare cases, a parking guidance system considers a communication model with multiple wireless sensor systems for covering a single parking lot, and the unification of the data happens in the cloud-based application [19]. Another parking management system considers hierarchical occupancy data collation where sensor nodes communicate to group nodes and group nodes report to a central control node [20]. The parking systems do not measure the network performance of the sensing layer irrespective of its communication model being a flat network or clusters inside a single network or multiple networks. This paper studies different communication models for the sensing layer and proposes finding an efficient model for data collation and dissemination.

Access technologies inter-connect various subsystems, and the performance of the whole system is directly related to the performance of the communication layer. IoT systems have many options for access technologies depending on the required range of the wireless communication, the network topology, data-sharing models, open technologies versus proprietary and the availability of standardization. The most important support for IoT's access technology came in the form of a tiny IPv6 stack for the low power-constrained devices designed by the standard body Internet Engineering Task Force (IETF) [21]. 6LoWPAN and RPL are standard messaging protocols that could increase the utilization of open-source components in building a dependable information and communication technology for a sustainable smart city [22].

RPL is a mesh-routing protocol that supports IPv6 addresses for IoT devices and is used extensively in smart utility networks and smart grids. There are a huge number of studies to measure the performance of RPL for data-gathering applications based on a variety of parameters. Studies conclude that the combination of utilizing ETX as the link metric and a radio duty cycling mechanism to synchronously turn on and off radios empowers RPL in terms of lowest energy consumption [23,24]. Similarly, the network topology is found to have an impact on RPL network's energy consumption, and a circular topology is found to more effective than a grid or tree [25]. A study finds that compressed sensing and data aggregation in RPL reduces the data latency as well as cuts down packet loss [26]. In contrast, Pham et al. shows the need for a scheduling mechanism for delivering the aggregated data packet to reduce the latency and proposes a novel relative collision graph algorithm-based scheduler [27].

Lim, in a survey paper, categorizes multiple sinks as a viable method to reduce congestion and improve RPL's performance in an IoT network [28]. Many research works propose to increase network performance by defining more than one instance of RPL under

a BR. Multi-sink approaches are proposed to handle high traffic volumes, offer safety against BR failure, combat congestion and balance traffic load across various forwarders [29–32]. A sink is a node that collates data; however, these works refer to the RPL root node as the sink.

The coordination between multiple sinks is proposed through a virtual root or through cooperative mechanisms between the different sinks [33–35]. Junior et al. argued that dynamism in invoking multiple instances is better than static multiple instances in handling different data traffic for multiple IoT applications [36]. Depending on the type of application, the node switches between stored instances to experience a reduction in control messages and power consumption. Hassani et al. show that combined metrics offer superior performance when compared to a single metric in a multi-sink scenario [37]. All such works incur modifications to RPL control messages, introduce new layers and increase the complexity. Furthermore, these works do not focus on exchanging data between multiple sinks, as they focus on a particular case of different sinks collating different type of data from the IoT network. Moreover, there is no need for the sink node to be the destination of data and any node in the network can act as a data server.

Tran et al. measure RPL's performance under different topologies such as linear, circular, random and grid. They conclude that the topology does not impact power consumption but influences latency [38]. The number of hops needed to reach the destination has an impact on the performance, as congestion is prevalent around the sink node. Hilmani et al. use a WSN for gathering occupancy data in the central gateway/sink node and apply a self-organizing algorithm for cluster formation [39]. In the clustering approach, there is no explicit insight on the exchange of data between clusters or the latency involved. Although there a plethora of works in the literature to improve the performance of RPL [40], the simple effects of the position of root node or the usage of multiple servers to collate data are not studied.

## 4. Efficient Communication Model for an SPS with Multiple Data Consumers

IoT applications implemented with low-power personal area networks have a variety of requirements such as low power consumption, low latency, less traffic overload and high reliability [41]. In order to satisfy these requirements, an efficient communication model must:

- Provide reliable data collection in a large mesh network;
- Minimize the power consumption of the battery-operated IoT devices;
- Be quicker in collating and furnishing the data to consumer devices;
- Have a data format that compresses the volume of data.

A parking lot application that generates parking availability data has to forge effective communication paths between IoT devices that generate occupancy data, devices that collate the occupancy data and devices that consume occupancy data. The performance of a multi-hop network depends on the number of hops between the source and the destination. When data are transmitted through a minimal number of nodes, the latency and power consumption are optimal. On this basis, five different communication models are evaluated for implementing an SPS with multiple data consumers:

1. Border router with a single data collator at the perimeter of the parking lot;
2. Border router with a single data collator at the center of the parking lot;
3. Border router with multiple data collators distributed across the parking lot;
4. Border router with four data collators at the center of the parking lot;
5. Border router with four data collators at the center and each forwarder in the mesh aggregates occupancy data.

A border router (BR) facilitates connections between the mesh devices and Internet backbone. A BR aggregates routes to all mesh nodes and utilizes the same to connect them with hosts from other IP-based networks [6]. The wireless connection between all these entities forms the communication ecosystem of the SPS. In order to realize the goals of an

efficient communication model, several aspects such as the position of border router and data collators, radio duty cycling, and data formats are considered in this work. Figure 6a–d represent a class of communication model where IoT devices simply forwards data toward the data collator. However, the position of the data collators vary among them. Except for the third model in Figure 6c, the BR and data collators are neighbors. The third model has split the entire network into four quadrants and has one data collator at the center of each quadrant. This places the data collator nested among the data producers. In contrast, Figure 6e shows a model where forwarders accept data from their children nodes, aggregate and then send out a single data packet with the consolidated occupancy data.

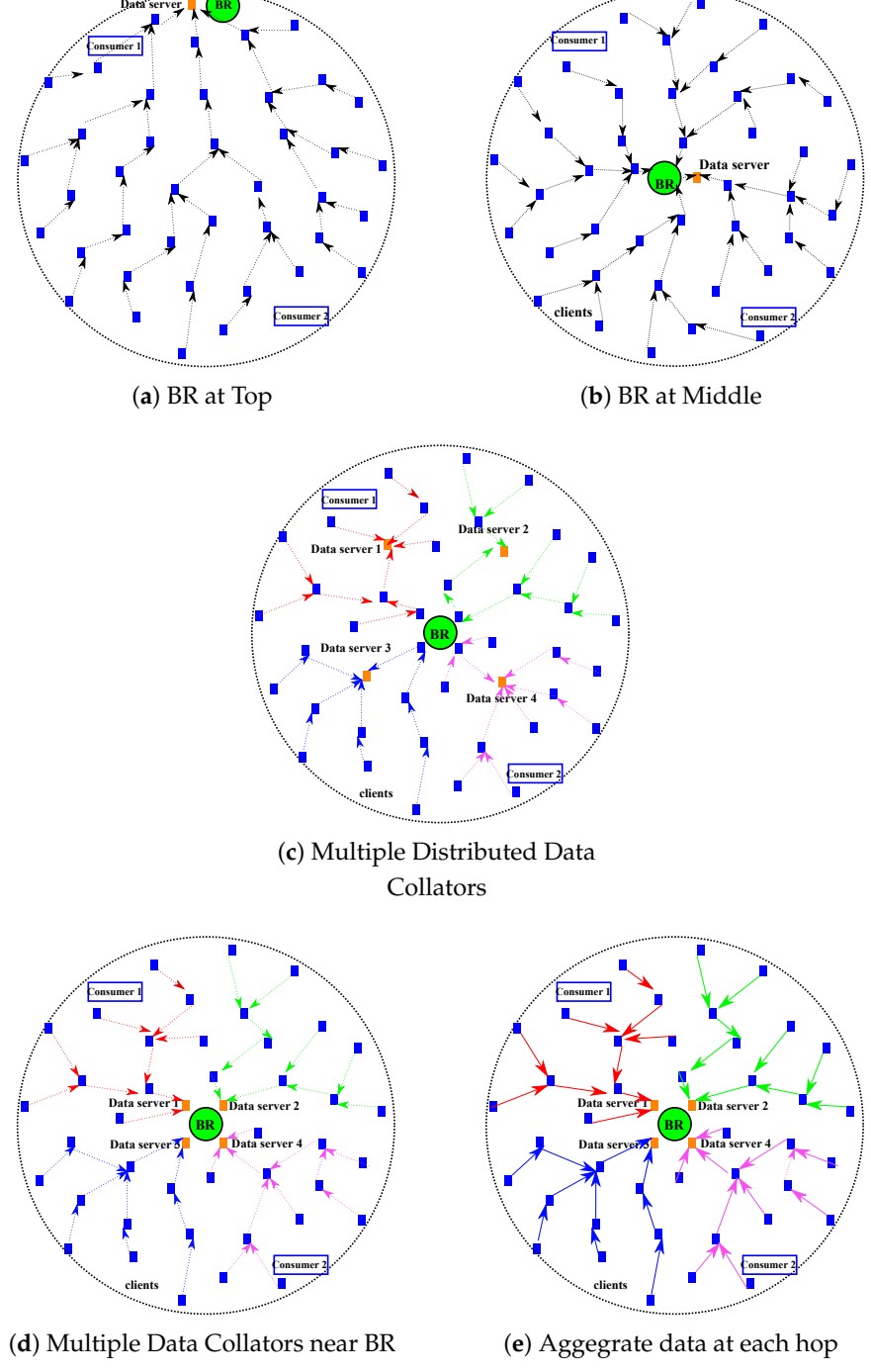

(**a**) BR at Top

(**b**) BR at Middle

(**c**) Multiple Distributed Data Collators

(**d**) Multiple Data Collators near BR

(**e**) Aggregate data at each hop

**Figure 6.** Flow of occupancy data (**a**) BR with one data collator at top. (**b**) BR with one data collator in middle (**c**) BR with four distributed data collators (**d**) BR with four data collators near BR. (**e**) Routers aggregate occupancy data at each hop.

The past research work on RPL's performance provides several pieces of vital information, including ETX for best link assessment, the significance of radio power consumption, duty cycling for reducing energy consumption, the relation between topology and performance and load balancing with multiple sinks. As multiple sinks mean more border routers, it involves high control overhead in maintaining more than one instance of RPL. Instead, this paper explores several alternate aspects such as multiple data collators, their positions, relative positions with data consumers, duty cycles of IoT devices, and data exchange format for arriving at a simple and efficient communication model.

### 4.1. Positioning BR for a Balanced Mesh Formation

BR is the root node in the mesh that initiates the mesh formation. RPL protocol forms a directed acyclic graph (DAG) that is destined to the BR by sending a DAG information object (DIO). DIO is an advertisement, and nodes hearing it join the DAG. It then furthers the transmission of DIO using trickle timers, and nodes join as in a ripple. Hence, nodes at the far end of the network perimeter takes more time to join the DAG. To have a uniform distribution of DIO along the perimeter of the network, it is necessary for the BR to be at the center of the network. This ensures that all nodes along the entire perimeter of the network have the smallest possible hops to reach the BR. With the number of hops directly proportional to the energy utilization and latency, positioning the BR at the center is the best approach. In addition, congestion around the BR node is quite low for a network having BR at the center when compared to a BR at the top (as in Figure 6a).

### 4.2. Positioning Data Collators for Reliable and Faster Data Collection

The BR itself can act as a data collecting point as RPL has a reliable DAG path to the root node. This introduces the funneling effect where forwarders close to the BR experience huge traffic. To reduce this effect, many researchers propose using multiple root nodes and collating the data outside the RPL network [42,43]. However, this deprives the on-site data consumers from directly accessing the occupancy data within the network and adds a dependency to the Internet connection besides increasing the delay in acquiring the data. The multi-sink approaches are complex with additional systems and modifications to the RPL control messages. As an alternate, multiple data servers are proposed in this work. It is essential that the data servers are stationed close to a BR so that a data server can reach another through the BR. This reduces the funneling effect and requires no complex improvisations to the RPL protocol. As the data servers are en route to BR, all the data-producing nodes already have an optimal path to reach the data collator. To illustrate this point, the third model has multiple data collators away from the BR.

### 4.3. Hybrid Power Cycling for Mesh Devices

Thread is an emerging routing protocol that is extensively used in smart home appliances [44]. The thread's communication model has mains-powered thread routers and duty-cycled sleepy end devices. Such a hybrid power cycling works for smart home mesh networks. However, the mains power is not suitable for mesh forwarders employed in a smart parking lot because a huge number of forwarders are required to cover the entire parking lot. However, the data collators are smaller in numbers and can be mains powered to maximize the packet reception rate of the data servers and also cater to high data volumes. They mains-powered devices do not switch off their radios. All other sensor nodes can have radio duty cycling to reduce energy consumption and switch off their radios most of the time except during packet transmissions. The BR, too, has its radio on so as to have a seamless connection to Internet. The positioning of BR makes it easier to extend the same to the data servers that are nearby. This hybrid power solution allows for lower energy consumption for the battery-powered nodes and ensures reliable occupancy data collection for the data servers.

*4.4. Concise Data Format for Data Exchange*

The occupancy data can be expressed in binary as they are two-state data, which are either occupied or not occupied. Hence, a single byte can represent eight parking slots and an 80-byte IP payload can effectively contain occupancy data for 640 parking slots. The occupancy data can be multiplexed at the data servers and is used for exchanging collated data between servers. The same is sent to data consumers. The concise string is then broken down back to occupancy data in the consumer node. The BR uses the compressed occupancy string in the HTTP data exchanged with the web browser. This short data exchange format reduces the load time of the web page.

The algorithm presented in Algorithm 1 takes the occupancy data array and converts the same to a single consolidated string. The index of the array is mapped to the position of the parking lot and is subsequently filled with either zero or one. The data collators fill the respective slots in the data array and convert the data to a concise string. This string is eight times compressed and can hold over 600 occupancy data in a single IP payload of IEEE 802.15.4 mesh.

---

**Algorithm 1** Convert occupancy data to a concise string.

---

Input is $ODA$, Occupancy data array
Output is *occupancy_str*, Concised occupancy data as string
**for** $i = 0$ to *no_of_parking_slots* $- 1$ **do**
    **for** $j = 0$ to $7$ **do**
        *shifted_bit* $= ODA[i] \ << \ j$
        *combined_byte* $= combined\_byte \ | \ shifted\_bit$
    **end for**
    *byte_str* $= int\_to\_char(combined\_byte)$
    concatenate *byte_str* with *occupancy_str*
**end for**

---

The algorithm presented in Algorithm 2 takes the consolidated string and converts it back to occupancy data.

---

**Algorithm 2** Convert the concise string back to occupancy data.

---

Input is *occupancy_str*, Concised occupancy data as array of characters
Output is $ODA$, Occupancy data array
*bit_mask*[] is {128, 64, 32, 16, 8, 4, 2, 1}
**for** $i = 0$ to *no_of_parking_slots* $- 1$ **do**
    *int_value* $= char\_to\_int(occupancy\_str[i])$
    **for** $j = 0$ to $7$ **do**
        *occupancy_bit* $= bit\_mask[j] \ \& \ int\_value$
        **if** *occupancy_bit* $> 0$ **then**
            *occupancy_bit* $= 1$
        **else**
            *occupancy_bit* $= 0$
        **end if**
        $ODA[i] = occupancy\_bit$
    **end for**
**end for**

---

All these decisions are expected to play a role in establishing efficient communication between all the concerned entities of the SPS system.

## 5. Evaluations

All the five different communication models referenced in the previous section are evaluated against each other for their efficiency in terms of data loss, packet latency, control overhead, energy consumption, time needed to obtain occupancy data for all the parking

slots and the time taken for the occupancy data to reach the consumers. To this effect, an experimental study is carried out in a simulated IPv6 mesh network with 100 nodes and one BR. The Cooja simulator is a widely accepted simulator for conducting experimental studies of IEEE 802.15.4 based IoT networks [45].

*5.1. Simulation Setup*

In the experiments, the BR is the root node of the IPv6 mesh and creates a DODAG with one RPL instance. All the nodes are forwarders that are capable of forwarding data packets in the upward direction toward the root node (BR). The data collator is placed at a one-hop position from the BR so that it lies in the upward path en route to the BR for each node. The BR is connected through SLIP protocol to a laptop. The Firefox browser is used from the laptop to connect to the data collator to access the occupancy data over the Internet. The five networks to be examined are labeled as Top1, Mid1, Dist4, Mid4 and MidAgg as per their communication model. The model named Top1 denotes a network with a single BR and one data collator placed at the top of all the nodes. Mid1 refers to the network with a single BR and one data collator at the center of the network. The third model, Dist4, has the BR at the center, and its four data collators are distributed within the network and are away from the BR. In contrast, Mid4 refers to four data collators that are adjacent to the BR, at the middle of the network. The final MidAgg model denotes a network with four data collators in the center where each node aggregates the occupancy data. The final model is expected to consume less energy, as it reduces the total number of occupancy data packets transmitted in the network.

The grid network is considered for simulation, as the results are comparable across multiple studies. The channel check rate for a node is kept at 8HZ so as to reduce the power consumption of the nodes. A radio duty cycling ensures that the nodes remain in sleep mode for as long as possible. The data collators do not participate in radio duty cycling to ensure high reliability. The simulation parameters are summarized in Table 1.

**Table 1.** Configuration parameters for the simulation study.

| Network Parameter | Value |
| --- | --- |
| Node placement | $10 \times 10$ uniform grid |
| Radio medium | UDGM |
| Distance between nodes | 30 m |
| TX Range/INT Range | 50 m/100 m |
| IoT devices having parking sensors | 96 |
| IoT devices as on-site data consumers | 2, top first node and bottom last node |
| Number of BR | 1 |
| BR position | Top or Center as per the models |
| Number of data collators | 1 or 4 depending on the model |
| Data collators position | Top/Center of quadrants/Center |
| Mode of operation | Storing mode |
| Run Time | 3600 s |
| Occupancy sensing interval | 60 s |
| Radio duty cycling for parking sensors | ContikiMAC |
| RDC for data collators and BR | None |
| Channel check rate | 8 HZ |

After an initial delay of 120 s, each parking sensor node generates a data packet with occupancy data every 60 s. The data are either 0 or 1, depending on whether the respective slot is vacant or occupied. The data packet is addressed to the data server whose address is sought through service discovery. In case of multiple data collators, the address of the first discovered server is considered, since it would be the most nearest server. The data collators exchange data once every 60 s between them and also send the collated data to on-site consumers.

*5.2. Simulation Results*

The nodes record the number of occupancy data packets dispatched, the time of packet transmission, the number of packets received, the arrival time of the data packet, the number of different control messages transmitted for setting the mesh and the duration for its radio being active. The packet delivery ratio (PDR) is measured as the percentage of the number of occupancy data packets received at the collator(s) to that of the number of packets sent. A high PDR indicates reliable communication between the nodes and the data collator(s). The graph in Figure 7a shows the PDR of all the four communication models. As expected, it is 98.2% for the Top1 model, which has one BR and one data collector positioned at the top of all of the nodes. The model shows some initial packet loss for nodes with longer paths. The longer a data path, the more time it takes to stabilize. Dist4 also exhibits some packet loss, as nodes take relatively longer routes to data collators. The data packets have to travel upwards along the DAG to a common ancestor and then downwards toward the data collator. As the network becomes bigger, both Top1 and Dist4 would experience a further increase in the data path length. The PDR for all the other three models are almost the same and report negligible packet loss.

Figure 7b presents the average number of control packets transmitted by a node. RPL uses three different control packets, DAG information solicitation message (DIS) and DAG information object message (DIO) for forming upward routes, and (Destination advertisement object (DAO) for forming downward routes [6]. Nodes are required to send control packets in order to create and maintain the fail-safe routing paths. Less control overhead reflects the efficiency of the multi-hop mesh creation process and conserves energy in a network. The Mid4 model keeps the control overhead lowest among the models, which is closely followed by the Mid1 model.

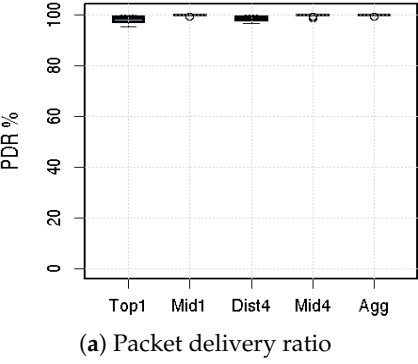

(**a**) Packet delivery ratio

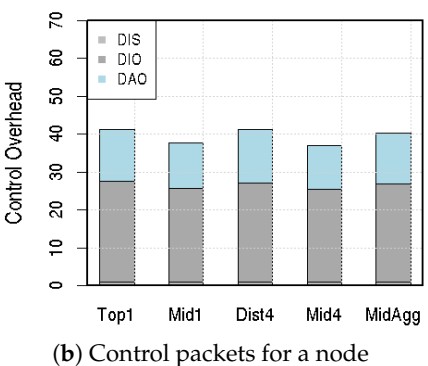

(**b**) Control packets for a node

**Figure 7.** Metrics for the communication models. (**a**) Data reliability in the network (**b**) Control overhead in the network.

The occupancy data packet latency is an average measure of the time duration for each data packet to reach the destination from its corresponding origin. Graph Figure 8a displays a 164.7 ms latency for MidAgg model and a comparable 471 ms and 420 ms for Mid1 and Mid4, respectively. The nodes in Dist4 model experience a latency of 823.2 ms even when there are multiple data collators. The higher latency reflects the longer routes along the DAG to the data collators. The packet latency is lowest in MidAgg because the occupancy data packets are not sent to the data collator but are sent to the immediate one-hop forwarder parent. Hence, the lowest average packet latency corresponds to one level of data aggregation. It must be noted that the occupancy data would take longer to reach the data collator as it has to cross multiple aggregation on its way.

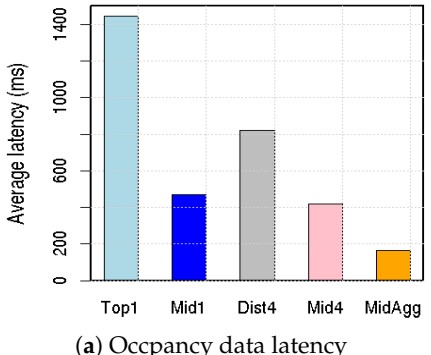
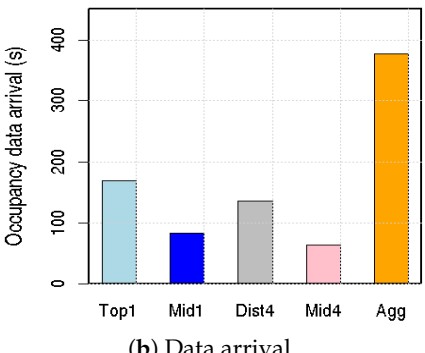

(**a**) Occpancy data latency

(**b**) Data arrival

**Figure 8.** Metrics for the communication models. (**a**) Latency for occupancy data packets. (**b**) Arrival of occupancy data from all nodes.

The next graph in Figure 8b shows the total time taken for the occupancy data from all the nodes in the network to reach the data collator. This is an important metric, as it shows the efficiency of the data collation in an SPS. The MidAgg model's under performance is because of the delay introduced by the aggregation at each hop. Both Mid1 amd Mid4 network's performances are lowest in the range of 83 s and 63 s, respectively.

The other metrics measured are the average packet latency for data packets between the data collator and the data consumer. Here, all the models have a similar delay under 1 s for one data consumer, but Top1 shows an elevated delay for one consumer, as shown in Figure 9a. The data consumers are placed at opposite sites of the network to simulate the presence of display screens at two far ends of a parking lot. So, when the data collator is at the top, it doubles the number of hops to reach a consumer at the far end. Dist4 exhibits a faster reach to consumers, since the distribution of data collators puts them closer to the consumer. This shows that the BR in middle is an efficient strategy to reach multiple consumers at the same time. Figure 9b visualizes the percentage of run time for which the radio was kept active. The first box shows the transmitter being active, and the second box shows the receiver active time. The transmitter is kept below 1% for models except the Top1 and Dist4, and receivers are kept active for less than 2%. Keeping the radios idle for longer helps conserve energy in the wireless network.

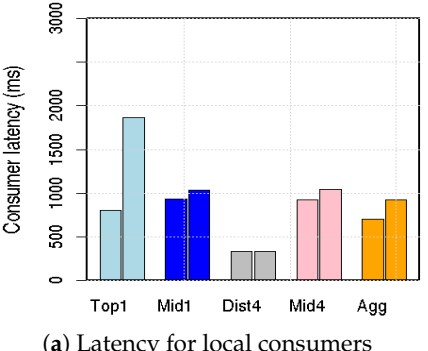
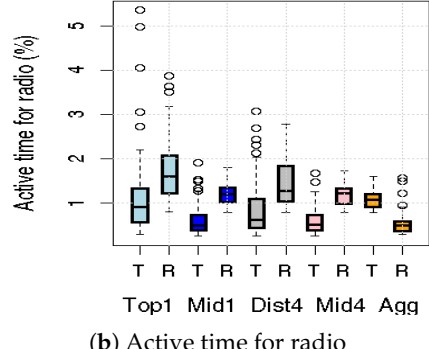

(**a**) Latency for local consumers

(**b**) Active time for radio

**Figure 9.** Metrics for the communication models. (**a**) Time to reach local consumers. (**b**) Percentage of time when radio was active.

In order to understand the energy utilization of nodes over time, the simulation is run for one hour with the energest metric report once every 5 min. The graph in Figure 10a showcases the average energy utilization of a node in different models. The initial spike is attributed to the network formation. There is a clear ranking in the energy consumption with Top1 being the highest with more packet transmission due to their longer distance to the BR. In the Dist4 model, energy utilization for a node is 22.7% higher than a node in the Mid1 or Mid4 models. Figure 10b displays the total charge consumed by a node in one

hour. This also confirms the earlier findings and denotes that the Mid1 and Mid4 models outperform others in terms of efficient data collation and dissemination.

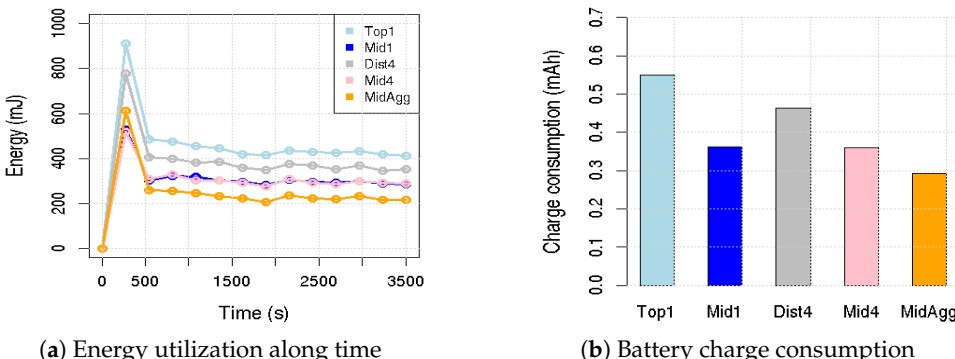

(**a**) Energy utilization along time    (**b**) Battery charge consumption

**Figure 10.** Metrics for the communication models. (**a**) Average energy utilization of a single node. (**b**) Battery charge consumption for one hour.

## 6. Results Discussion

When comparing the performances of the five different communication models, it is evident that the Mid1 and Mid4 models are showing good packet delivery ratio along with low overhead. Although the MidAgg model exhibits a low packet latency along with the lowest power consumption, the time taken for the occupancy data to reach the data collator is six times over the time taken for the Mid4 model or five times over the time taken for the Mid1 model. It requires a specific scheduling algorithm for packet transmission that can reduce the delay introduced by data aggregation at each hop. The Top1 model apparently demonstrates a lower data reliability and a high packet latency as the data packets need to traverse a higher number of hops than the other models. Between these Mid1 and Mid4 models, the Mid1's packet delivery ratio has an edge over Mid4. However, latency wise, the Mid4 model holds an edge. To understand the advantage of these two models, the experiments are repeated in a larger 15 × 15 grid network.

*Assessing the Scalability of Mid1 and Mid4 Models*

The graph in Figure 11a presents the PDR for Mid1 and Mid4 models in a 10 × 10 grid network against the 15 × 15 grid network.

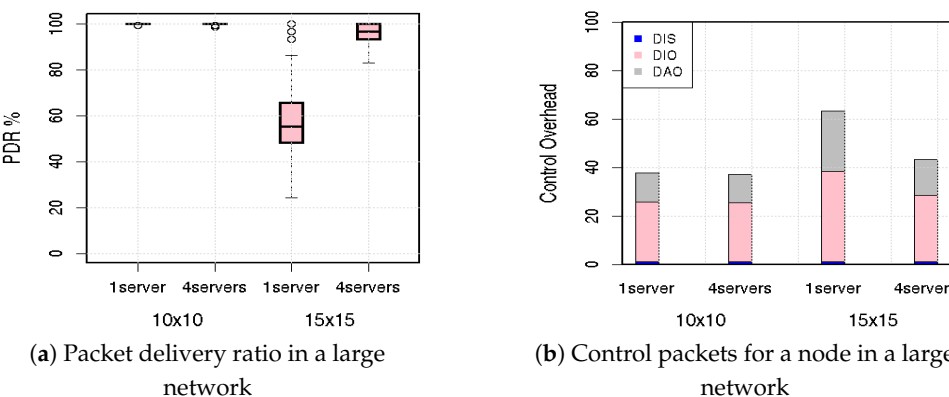

(**a**) Packet delivery ratio in a large
network

(**b**) Control packets for a node in a large
network

**Figure 11.** Performance in a 10 × 10 grid vs. 15 × 15 grid. (**a**) Data reliability in the network. (**b**) Control overhead in the network.

In a larger network, the differences between the two models are evident. The Mid4 model outperforms and is 38.1% more efficient than the Mid1 model toward reliable data collation. Figure 11b shows a small rise in control packets for the Mid4 model in a scaled-up network. However, Mid1 suffers from a 40.9% increase in control overhead when compared to the same model in a smaller network.

The data latency metrics for the two models are presented in Figure 12. The average time taken by the occupancy data packet latency to reach the data collator is very high, clocking over 12 s for the Mid1 model. Mid4 takes about 2 s for reaching the data collators and shows a clear superior performance. As the number of nodes in the network increases, the congestion causes a severe funneling effect around the root node. Hence, the performance of the Mid1 model is very low in a large network.

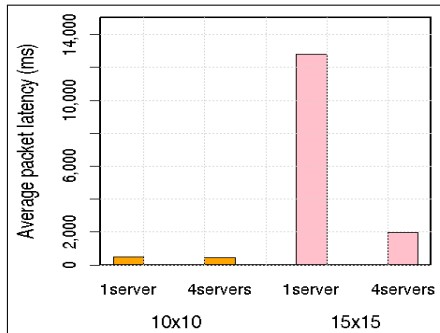

**Figure 12.** Occpancy data latency in a large network.

A similar trend is shown in Figure 13a for the time taken to reach the data consumers, and Mid4 outperforms the Mid1 model. The energy consumption is also lower for the Mid4 model, and the same is illustrated in Figure 13b. It can be concluded that a multi-data collator model with the BR at the center of the network fits the efficient communication model requirement for an SPS.

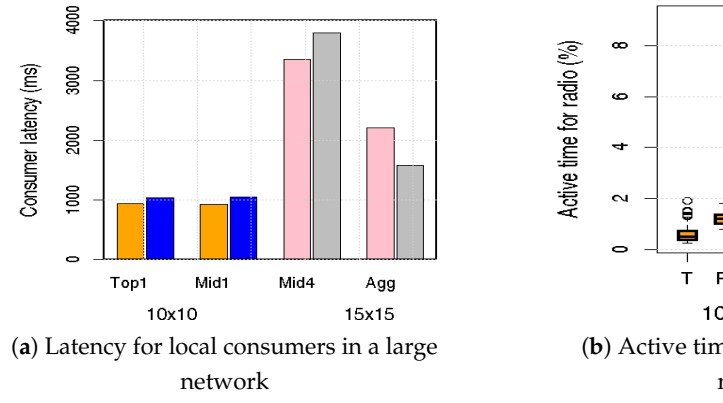

(**a**) Latency for local consumers in a large network

(**b**) Active time for radio in a large network

**Figure 13.** Performance in a 10 × 10 grid vs. 15 × 15 grid (**a**) Time to reach local consumers. (**b**) Percentage of time when radio was active.

## 7. Conclusions

The communication technology is a vital component of an SPS system, and it is necessary to have an effective communication model that provides reliable and faster occupancy data collation and dissemination between different entities. This paper explored various aspects such as the position of the BR in a mesh network, the presence of a single data collator against multiple data collators, their relative positions with respect to BR, consumers and the effects of hybrid radio duty cycling for mesh devices. It also proposed a concise data format that accommodates a large number of occupancy data (up to 640 parking slots) in a single data packet. This reduces the number of data packets exchanged between the data collators and data consumers. Lowering the radio activity directly improves the energy efficiency of the system. Along with that, the concise data format presents a short http message and improves the web page load time. Five different communication models are evaluated for their efficiency in providing low latency and energy efficient communication. The best two models were further subjected to a scalability test in a larger 15 × 15 grid network. A multiple data collator model where the data

collators are adjacent to the BR and are positioned at the center of the network is identified as the best model for providing efficient communication between data producers and consumers. Having multiple data collators adjacent to BR reduces congestion around the BR in a large network and improves their reliability. Their position at a center point reduces the hop distance between the nodes and reduces latency. Congestion avoidance and shorter communication paths present an energy-efficient system. Thus, the strategic positioning of multiple data collators reduces data transit time, offers a higher data reliability and lowers the power consumption of the mesh devices.

**Author Contributions:** Conceptualization, T.A.; methodology, T.A. and M.P.; software, T.A.; validation, T.A. and M.P.; formal analysis, M.P. and T.A.; investigation, T.A.; resources, M.P.; data curation, T.A.; writing—original draft preparation, T.A.; writing—review and editing, M.P. and T.A.; visualization, T.A.; supervision, M.P.; project administration, M.P.; funding acquisition, Not Applicable. All authors have read and agreed to the published version of the manuscript.

**Funding:** This research received no external funding.

**Data Availability Statement:** The data are available with the corresponding author and can be provided on request.

**Conflicts of Interest:** The authors declare no conflict of interest.

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
