# Peer review of "Efficient Communication Model for a Smart Parking System with Multiple Data Consumers"

_smartcities, doi:10.3390/smartcities5040078_

Round 1

Reviewer 1 Report

The authors present a study to determine where to place the elements involved in a communication with the RPL protocol, especially the border router. The simulations are very simple and some of the conclusions are obvious. Some references could be added to the text, especially about the simulator used which is very old. English sounds good, the text is easy to read. The structure of the paper is adequate.

Some improvements that could be added to the paper are listed below.

1. References to more modern technologies such as 5G for IoT could be included in the introduction, which would efficiently serve the problem studied by the authors.

2. Lines 45-46. The bandwidth that would be used in the presented proposal would be very small, nor would there be any processing problem.

3. Line 62: speed of vehicles. I don't know of any car park where the speed of vehicles is measured.

4. Line 131. What is a serial protocol used for?

5. BR is not defined.

6. Section 4.1 Placing the BR in the center is obvious. The exact position of the aggregators is not indicated. If they are located in the center of the four possible quadrants, it would be very effective.

7. Algorithms 1 and 2 do not contribute anything and are trivial.

8. Table 1. Many data would be missing to be able to replicate the model.

9. Line 302. Why has this simulator been used? It's very old. Missing a reference about where it can be downloaded.

10. Section 5.2. Although the simulations do not contemplate it, the charge of the batteries and how it decreases over time would be an interesting metric to use.

11. Figure 8a. I don't understand why packets are lost in the Top 1 case.

12. Figure 8b. what is DIO, DIS and DAO? If the same IPv6 model is used in all the simulations, the overhead should be similar as shown in this figure.

13. Figure 9. That a data takes 1400 ms could be considered high, in any case, this value for the chosen parking application would not be a problem either.

14. Section 6. Simulations could be done with 50,100, 200, 300 nodes... The numbre 225 is a bit strange.

Reviewer 2 Report

The paper discusses an interesting ENT modern topic on smart parking systems. The paper is within the scope of the journal. The paper is generally evaluated well; however, it has some flaws.

·       Lines 76-87 describe SPS, but there are no references. Please add the sources of definitions, etc.

·       Figures should be first mentioned in the text and then placed in the paper. Correct it everywhere on the paper.

·       Figure 1 and Figure 2 should be one figure since Figure 2 explains Figure 1. Figure 2 is a type of description that should be added to Figure 1.

·       Table 1 is not done according to the template. Correct it please.

·       Are all the figures yours or are they taken from some other paper? if they are not yours, please add this source / reference. Also, make sure that you have copyright permission.

·       Correct lines 209 to 213 according to the template. Do not use a comma but a colon. Where do the assumptions come from? please justify it in text.

·       When using bullet points, please do it uniformly throughout the text. Do not use a comma but a colon. Why are these four different communication models considered for the implementation of an SPS? please justify it in the text.

·       I suggest adding a short paragraph in the literature review on models for a smart parking system with some references to literature, of course.

·       The algorithms 1, 2... should be mentioned in text before they appear.

·       Table 1 - please add the source of the parameters. justify the parameters.

·       The conclusions must be improved.  They are not of the best quality. I advise against placing drawings in conclusions.

Round 2

Reviewer 1 Report

The authors have reasonably responded to the petitions presented.

Reviewer 2 Report

Thank you for the improvements.